



# The recent developments in spatio-temporally continuous snow cover product generation

Xinghua Li[1], Yinghong Jing[2], Huanfeng Shen[2], Liangpei Zhang[3]

[1]School of Remote Sensing and Information Engineering, Wuhan University, China
[2]School of Resource and Environmental Sciences, Wuhan University, China
[3]State Key Laboratory of Information Engineering in Surveying, Mapping, and Remote Sensing, Wuhan University, China

*Correspondence to*: Huanfeng Shen (shenhf@whu.edu.cn)

**Abstract.** The snow cover products of optical remote sensing systems play an important role in research into global climate
change, the hydrological cycle, the energy balance, and so on. However, cloud cover results in spatial and temporal discontinuity for long-term snow monitoring. In the last few decades, a large number of cloud removal methods for snow cover products have been proposed. In this paper, our goal is to make a comprehensive summarization of the existing algorithms for generating spatio-temporally continuous snow cover products, and to expose the development trends. The methods of generating spatio-temporally continuous snow cover products are classified into spatial methods, temporal
methods, spatio-temporal methods, and multi-source fusion methods. Experiments were conducted to validate the reconstruction effect of the representative methods.

## 1. Introduction

Because of the high albedo, high thermal emissivity, low thermal conductivity, and water storage ability (Tait et al., 2000;Tekeli and Tekeli, 2012), snow has a significant influence on the energy balance (Robinson et al., 1993;Crawford et al.,
2013), the hydrological cycle (Şorman et al., 2007;Kostadinov and Lookingbill, 2015), and climate change (Cohen and Entekhabi, 1999;Brown, 2000). In recent years, more and more attention has been focused on monitoring the spatial and temporal change of snow cover (Gao et al., 2012). The traditional in-situ snow monitoring approach is conducted only sparsely, and is limited due to the large gaps in both space and time (Brown and Braaten, 1998). In contrast, remote sensing data have the advantages of a wide coverage area and short revisit period (Lopez et al., 2008;Zeng et al., 2013), and have
been an effective and alternative supplement for in-situ data since April 1960 through the Television Infrared Observation Satellite (TIROS) (Singer and Popham, 1963). According to the data source, remote sensing based snow cover products mainly include microwave-based products, combined products, and optical-based products, (Frei et al., 2012), as shown in Figure 1.





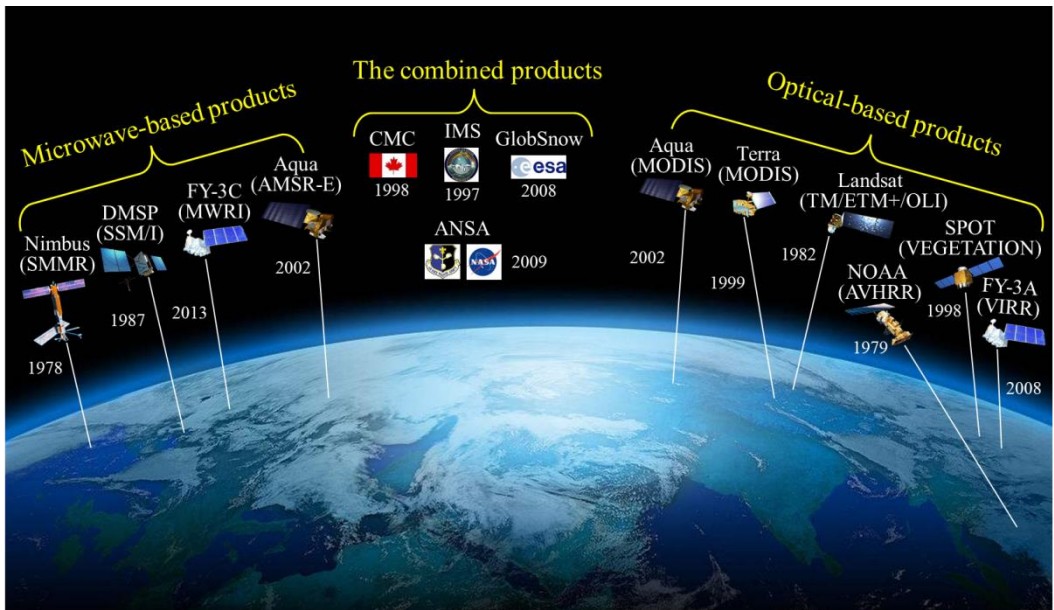

**Figure 1**. **Remote sensing based snow cover products.**

Microwave-based products are derived according to the relationship between microwave energy and snow depth (SD), or snow water equivalent (SWE) when the snowpack is dry (Tait et al., 2000;Wulder et al., 2007). Microwave-based products are free from cloud cover contamination and can capture the snow information with an all-time and all-weather ability. Since microwave penetrates most of the snow cover, it is possible to detect the SD and SWE. The typical products include the Scanning Multichannel Microwave Radiometer (SMMR) SD product (Chang et al., 1987), the Special Sensor Microwave/Imager (SSM/I) SD product (Grody and Basist, 1996), the Microwave Radiation Imager (MWRI) SD product (Che et al., 2016), and the Advanced Microwave Scanning Radiometer-Earth Observing System (AMSR-E) SWE product (Gao et al., 2010a;Anthony et al., 2008). Microwave-based products, which include both active and passive forms, have a high temporal resolution and can quickly cover the Earth's surface. Active microwave remote sensing has a high spatial resolution and can provide the details of snow cover. However, active microwave remote sensing is rarely used in research into SD because it features a highly transparent frequency to snow (Baghdadi et al., 1997) and can only recognize wet snow reliably (Dietz et al., 2012a). In contrast, passive microwave remote sensing is effective in retrieving both SD and SWE. However, the spatial resolution of passive microwave remote sensing systems is usually so low that they are not able to acquire the detailed information of snow (Foster et al., 1984). Generally speaking, passive microwave-based products are not suitable for meso/micro-scale research and may result in confusion between wet snow, thin snow, and forest (Tait et al., 2000). In addition, because of the limitations of the imaging orbit, microwave-based products are subject to spatial gaps.

Compared to the microwave-based products, optical-based products have no spatial gaps resulted from the imaging orbit. In order to derive spatio-temporally continuous snow cover products, combined products are generated, which are



combinations of satellite data (optical and microwave), climate station observations, and models (Frei et al., 2012;Tait et al., 2000). For example, the Canadian Meteorological Center (CMC) snow product (Drusch et al., 2004) is a combination of station observations and models, and GlobSnow (Metsämäki et al., 2015), from the European Space Agency (ESA), is a multiple-dataset snow cover product generated from satellite data, station observations, and models. In addition, the

Interactive Multisensor Snow and Ice Mapping System (IMS) produced by the U.S. National Ice Center (NIC) (Ramsay, 1998) is the fusion of many kinds of optical and microwave data, including Advanced Very High Resolution Radiometer (AVHRR) data, SSM/I data, AMSR-E data, Geostationary Operational Environmental Satellite (GOES) data, Polar Operational Environmental Satellite (POES) data, European Geostationary Meteorological Satellite (METEOSAT) data, Japanese Geostationary Meteorological Satellite (GMS) data, the National Centers for Environmental Prediction (NCEP)

model, U.S. Air Force (USAF) snow/ice analysis, and so on. The IMS is also jointly supported by the U.S. National Oceanic and Atmospheric Administration (NOAA), the U.S. Navy, and the U.S. Coast Guard. Regardless of cloud cover, the IMS produces near real-time products with spatial resolutions of ~1km, ~4km and ~24km, which provide the input for atmospheric forecast models (Brown et al., 2014). In addition, the Air Force Weather Agency (AFWA)/National Aeronautics and Space Administration (NASA) snow algorithm (ANSA) blends AMSR-E, Moderate Resolution Imaging

Spectroradiometer (MODIS) and Quick Scatterometer (QuikSCAT) data products (Foster et al., 2011). Since ANSA integrates optical, passive, and active microwave data, it can map the snow covered area (SCA), the fractional snow cover (FSC), the SWE, and the snowmelt area. The combined products draw together the respective advantages of each of the component products to improve the accuracy, quality, and spatio-temporal continuity of snow cover. Several problems encountered when the component products are used alone have been solved. The combined products provide more

information on the state of snow cover than each of the component products. However, the primary disadvantage of the combined products is the poor spatial resolution (Tait et al., 2000).

    Although the combined products have the advantage of spatio-temporal continuity, they need many different data sources, and their spatial resolution is restricted to the lowest spatial resolution among the data sources. As a result, many attempts have been made to derive spatio-temporally continuous snow cover products from optical remote sensing data. Optical-

based products have a higher spatial resolution than the combined products. Optical-based products are derived according to the differences in visible and infrared spectra between snow and cloud, bare land, vegetation, and so on. The representative products are mainly derived from the MODIS sensor onboard the Aqua and Terra satellites (Hall et al., 2002), AVHRR data (Simpson et al., 1998), VEGETATION data (Xiao et al., 2004), Thematic Mapper (TM) data (Rosenthal and Dozier, 1996)/Enhanced Thematic Mapper Plus (ETM+) data (McFadden et al., 2011)/Operational Land Imager (OLI) data

(Crawford, 2015), and Visible and Infrared Radiometer (VIRR) data (Zhang et al., 2017). Moreover, the weekly snow cover product (Brown et al., 2010) from NOAA is also derived from optical remote sensing imagery. The optical-based products have formed long observation time series and have a high spatial resolution. The unique spectral signature (e.g., high albedo, high thermal emissivity) of snow derived from optical observations is commonly used in the snow cover mapping algorithms (Simic et al., 2004). Optical observations can also be used to extract useful snow information. For



example, the SCA or snow cover extent (SCE) is usually extracted by binary classification or the use of the subpixel snow-fraction algorithm with the optical observations (Rittger et al., 2013;Salomonson and Appel, 2004). The optical-based products can not only be applied for monitoring the spatio-temporal variation of snow cover, but can also be set as the input of atmospheric forecast and hydrological models. As a result, the optical-based products are widely used in practical applications. However, they are inevitably subject to cloud cover contamination. Especially in the snow period, the cloud fraction (CF) of snow product is usually more than 10% (Liang et al., 2008b).

Among the existing optical-based snow cover products, the MODIS products have become one of the main data sources for ice and snow research. The MODIS products have advantages in the spatial and temporal resolutions, the global coverage, the long time series, and the free access, which together allow real-time, accurate, and large-area snow cover variation monitoring. The MODIS products are available through the National Snow and Ice Data Center Distributed Active Archive Center (NSIDC DAAC). The snow cover products of MODIS are derived from the SNOMAP algorithm (Hall et al., 1995). SNOMAP automatically uses the normalized difference snow index (NDSI) and decision strategies to identify snow cover (Hall et al., 2002), which makes this type of snow products more consistent with station observation data and other higher spatial resolution remote sensing data (Parajka and Blöschl, 2006;Wang et al., 2008;Bitner et al., 2002;Klein and Barnett, 2003;Chelamallu et al., 2014;Huang et al., 2011). According to accuracy assessments, under clear-sky conditions the overall accuracy (OA) of MODIS snow cover products is ranging from 85% to 99% (Parajka et al., 2012), and the overall absolute accuracy is ~93% (Hall and Riggs, 2007). Version 5 of MODIS snow cover products provides a binary product and a fractional snow product (Rittger et al., 2013;Li et al., 2018). The binary product is SCA and the fractional snow product is FSC, which are both derived from the NDSI (Singh et al., 2014;Liang et al., 2017). SCA can be used to estimate the regional snow resource when combined with SWE, and FSC is a very important parameter for estimating SWE (Molotch and Margulis, 2008). The MODIS products have temporal resolutions of 1 to 8 days and one month, and spatial resolutions from 500m to 0.05° (Déry et al., 2005). Like other optical-based products, the MODIS products are prone to contamination by a large percentage of cloud cover (Wang et al., 2009;Maskey et al., 2011), which limits the continuity of snow cover monitoring in space and time. Cloud cover also causes confusion in snow discrimination (Ault et al., 2006). As a result, it is of importance to remove the cloud coverage from the MODIS snow cover products.

In the past decades, there have been a large number of algorithms developed to improve the spatio-temporal continuity of optical-based snow cover products. The majority of the methods focus on the MODIS SCA products. The SCA products are usually just classification data, so the cloud removal method is notably different from the methods for traditional remote sensing images (Li et al., 2016;Shen et al., 2015;Li et al., 2014b). In this review, we comment on the recent developments in producing cloud-free MODIS snow cover products (without any special instructions, the product is MODIS SCA). The algorithms for generating spatio-temporally continuous snow cover products are mainly categorized into temporal methods, spatial methods, spatio-temporal methods, and multi-source fusion methods.

The remainder parts of this paper are arranged as the following. Section 2 introduces the spatial methods for generating spatio-temporally continuous snow cover products. The temporal methods, spatio-temporal methods, and multi-source fusion





methods are then introduced in Sections 3–5, respectively. Section 6 summarizes the current validations and evaluations of the spatio-temporally continuous snow cover products. Section 7 exposes the future direction. Finally, the conclusions are provided in Section 8.

## 2. Spatial methods

The spatial methods remove the cloud cover of the snow product based on the spatial distribution property of the snowpack. The main spatial methods are the spatial filter (SF), the snowline mapping approach (SNOWL), and the locally weighted logistic regression (LWLR).

### 2.1 The spatial filter (SF)

The most common spatial method is the SF (Gafurov and Bárdossy, 2009;Parajka and Blöschl, 2008;Paudel and Andersen,
2011), in which the idea is to replace a cloudy pixel using the four or eight neighboring non-cloud pixels. There have been many rules put forward to replace the cloudy pixel, as follows: (1) The cloud pixel should be reassigned as a snow pixel on the condition that three of its four direct "side-bordering" neighboring pixels are snow (Paudel and Andersen, 2011;Lindsay et al., 2015); (2) the cloudy pixel is replaced by the main classification (land or snow), i.e., the class of the majority of the non-cloud pixels in a neighborhood is used to replace the cloudy pixel (when there is a tie, the cloudy pixel is replaced by
snow) (Parajka and Blöschl, 2008;Tong et al., 2009a, b); (3) Gafurov and Bárdossy (2009) proposed that when the eight neighboring pixels with a lower elevation than the cloudy pixel are covered by snow, the cloudy pixel should be assigned as being snow-covered; and (4) López-Burgos et al. (2013) replaced the cloudy pixel based on the elevation and aspect; that is, if any neighboring pixel has snow cover, and has the same aspect and a lower elevation, then the cloudy pixel is classified as snow. For example, based on Rule (1), the cloudy pixel (marked as red star) in Figure 2a will be assigned as snow, and based
on Rule (2), the cloudy pixel in Figure 2b will be assigned as no snow. However, based on Rule (3), the cloudy pixel in Figure 2b will be assigned as snow when the snow pixels have lower elevations than the cloudy pixel. In this case, the most suitable rule should be chosen according to the regional snow cover change rule.

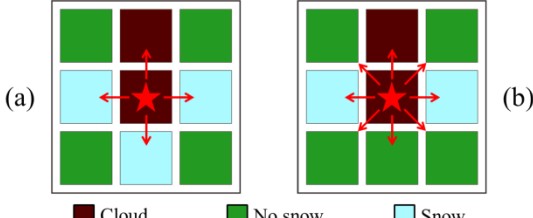

**Figure 2. Spatial filter. (a) Four-pixel neighborhood. (b) Eight-pixel neighborhood.**

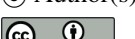



The SF is usually effective for small-area cloud cover, and cloud with a proportion of no more than 10% can be removed. For example, the SF in an eight-neighborhood decreases the cloud coverage of MODIS products by 7%, and the decrease in OA is just 0.7% (Parajka and Blöschl, 2008). According to practical application, the SF is not very sensitive to the size of filter window (Zhou et al., 2005). Additionally, in mountainous regions, the elevation is assumed to the dominant factor
affecting the snow cover distribution. Due to the complicated topography, the accuracy of the SF usually declines with elevation rising (Tong et al., 2009b).

## 2.2 The snowline mapping approach (SNOWL)

The SNOWL method (Parajka et al., 2010;Dietz et al., 2013), which is also called the snow transition elevation method (Gafurov and Bárdossy, 2009;Shea et al., 2013), reclassifies the cloudy pixels as snow or land according to the elevation
distribution characteristics of the snowpack. The cornerstone of SNOWL is the snowline and the landline. As shown in Figure 3, the snowline is the snow-covered elevation where all pixels above it are covered by snow, and the landline is the minimum elevation where snow exists (Krajčí et al., 2014;Krajčí et al., 2016;Lei et al., 2012). On that account, all the cloudy pixels above the snowline are assigned as snow, and all the cloudy pixels below the landline are assigned as land by SNOWL. The cloudy pixels between the snowline and landline are assigned as partial snow. However, the partial snow brings some
uncertainties to the monitoring of snow cover variation.

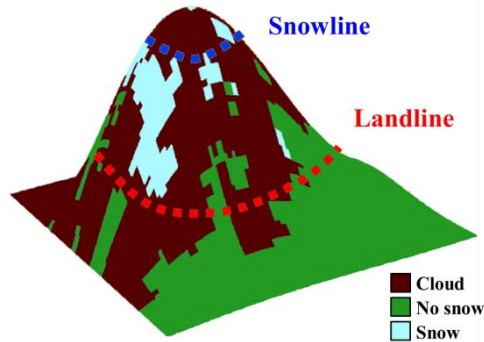

**Figure 3. Snowline and landline.**

SNOWL is relatively simple and easy to carry out, and performs well in both high and low elevations. In order to make sure that the snowline and landline are accurately assigned, the product should be at least 70% cloud free (Gafurov and Bárdossy, 2009). Generally speaking, as the CF increases, the accuracy of cloud removal decreases. Some scholars considered aspect, topography, and land cover classes to improve the accuracy of SNOWL (Paudel and Andersen, 2011;Da Ronco and De Michele, 2014a). In a few special cases, the snowline for the whole area is not met, so the regional snowline
for the local area is assigned (Parajka et al., 2010). In the Trans-Himalayan region, the improved regional SNOWL method removes 38.28% of the cloud, and the commission error is less than 2% (Paudel and Andersen, 2011).



### 2.3 Locally weighted logistic regression (LWLR)

For LWLR method (López-Burgos et al., 2013), the snow cover probability of the cloudy pixel is predicted by the topographic and spatial relations with its neighboring cloudless pixels. LWLR enforces on two explanatory variables of elevation and aspect for snow occurrence probability. The information of the neighboring pixels is inversely weighted with

distance and is fit to a logistic curve. The pixels closer to the cloudy pixel are assigned with a heavier weight than those that are farther apart. Finally, the estimated snow probability by LWLR is converted to a binary result according to a selected threshold. However, how to choose a better threshold to obtain a binary result needs more tests, and its high cost limits its application, to some degree. Experiments demonstrated that the LWLR method could reduce the cloud obscuration by 93.8% in the Salt River basin in Arizona (López-Burgos et al., 2013).

Among all the spatial methods, the computational complexity of LWLR is the highest. The spatial methods mainly depend on the spatial distribution property of the snowpack to reclassify the cloudy pixels. For the majority of the spatial methods, the core idea is to utilize neighboring cloud-free pixels. However, when the CF is high, the accuracy will decrease.

### 3. Temporal methods

The temporal methods reclassify the cloudy pixels according to the temporal correlation and change rule of the snow cover.

According to the time span of the product used, the temporal methods are mainly classified as the Terra and Aqua combined (TAC) method and temporal filters. The reason why the TAC method is considered as a kind of temporal method, in our opinion, other than the multi-source fusion, is that the Aqua and Terra satellites are both equipped with MODIS (of nearly the same design). Their combination amounts to a composite of MODIS at different times.

### 3.1 Terra and Aqua combined (TAC)

Among the temporal methods, the TAC method is the simplest and most straightforward method. Since cloud coverage is changeful in a few hours, the TAC method (Parajka and Blöschl, 2008;Xie et al., 2009;Wang and Xie, 2009;Mazari et al., 2013) has become the most popular temporal method. The TAC method can decrease the cloud coverage ratio by 5–20%, without sacrificing obvious accuracy of the products. This method blends the same-day MODIS snow products on a pixel basis. As shown in Figure 4, if a pixel is cloudy in one product and cloud-free in another product, the cloudy pixel will be

updated by the classification of the cloud-free pixel. The combination usually has the following priority scheme: snow>no snow>cloud.



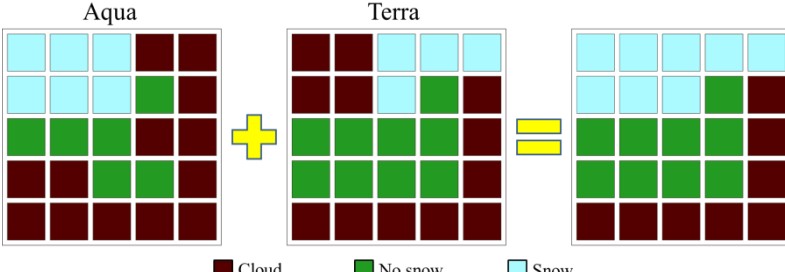

**Figure 4. Terra and Aqua combined.**

The basic assumption of the TAC method is that snowmelt and snowfall did not occur during the time interval. In the process of merging the two products, they are considered to be identical. In fact, the Terra and Aqua products still have some small differences, for the following reasons. The first is that they are acquired at different times (three hours). The second is that most of the detectors in Aqua MODIS band 6 failed (Shen et al., 2014). In the early days, the snow mapping method for Aqua used band 7 as a substitute for band 6; later on, Aqua MODIS band 6 was restored by quantitative image restoration method (Gladkova et al., 2012) with a high degree of accuracy. Even so, the two products still have slight differences under some conditions. According to the ground observations, the TAC method inherits the omission and commission errors of the input data of Terra and Aqua (Xia et al., 2012). Gao et al. (2010b) confirmed that the TAC method can reduce the cloud cover by 5–14% and 8–12% at the yearly and monthly scale, respectively, and the OA is 89.7%, which is lower than MOD10A1 by 0.7% and higher than MYD10A1 by 1.4%.

### 3.2 Temporal filters

Another popular kind of temporal method is the temporal filter methods (Parajka et al., 2012;Hori et al., 2017), which mainly include adjacent temporal deduction (ATD), multi-day combination (MDC), season filter (SFil), and temporal interpolation using a mathematical function. The first three kinds of methods are applied to the SCA product (as shown in Figure 5), and the last method is suitable for the FSC product. It should be noted that temporal filters are also referred to as temporal interpolation methods in some literature.




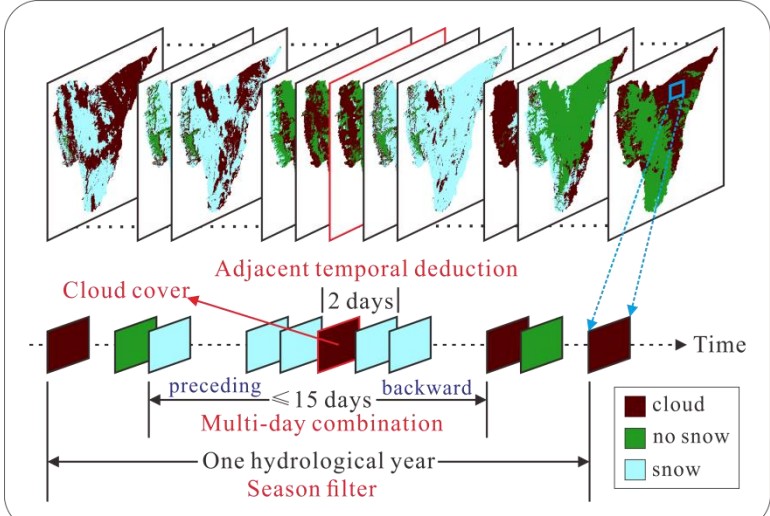

**Figure 5. Temporal filters for the MODIS SCA products.**

### 3.2.1 Adjacent temporal deduction (ATD)

ATD (Paudel and Andersen, 2011;Lindsay et al., 2015;Dietz et al., 2013) is an effective way to deduce the surface conditions via the same pixel in the previous and subsequent days, without reducing the spatial and temporal resolution. It is assumed that if the preceding and the following day of the cloudy pixel remain the same (land or snow), the condition of the cloudy pixel will remain unchanged (Gafurov and Bárdossy, 2009). When the previous and the next day are different, the cloudy pixel is still assigned as cloud. In fact, ATD uses ±1-day information, and is similar to the works of Li et al. (2008)

and Gafurov and Bárdossy (2009), who used ±1-day or ±2-day information. However, ATD is different from direct temporal replacement (Zhao and Fernandes, 2009), with the sacrifice of the 1-day temporal resolution.

Owing to the useful information from the previous day and the subsequent day, the accuracy of cloudy pixel reclassification is high. On the basis of the TAC method, ATD can not only decrease CF by 25%, but also achieve an accuracy of 96.3% (Gafurov and Bárdossy, 2009). However, since the snow cover is assumed to remain unchanged in the

15 given temporal interval, the accuracy in snow-transitional periods is obviously lower than that in snow-stable periods (Gao et al., 2010b). In other words, ATD is not suitable for an operational context.

### 3.2.2 Multi-day combination (MDC)

Among the temporal filter methods, MDC (Parajka and Blöschl, 2008;Dietz et al., 2012b;Zhang et al., 2012;Gao et al., 2011a) is the most widely used method. The cloudy pixels are replaced by the cloudless pixels in a temporal interval ranging

from one to more than 10 days with a constant or flexible way (Chen et al., 2014). To some degree, the above-mentioned ATD can be regarded as a special case of MDC. As far as MDC is concerned, the temporal window can be preceding (Wang



et al., 2014) or preceding and backward (Sharma et al., 2014;Foppa and Seiz, 2012;Coll and Li, 2018). Usually, priority is given to the preceding days (Marchane et al., 2015). For the preceding temporal window, it amounts to a backward temporal filter, in which the cloudy pixel is replaced by the previous cloud-free pixel in turn. The cloud-gap-filled (CGF) (Hall et al., 2010) method also belongs to this scenario. When the land cover is considered to be unchanged, the information of the two days forward and the two days backward can be used (Gafurov and Bárdossy, 2009;Da Ronco and De Michele, 2014a). To improve the accuracy of the cloud removal, certain thresholds (e.g., cloud percentage and the number of composite days) should be imposed in MDC (Gao et al., 2010b).

   MDC can reduce a high fraction of cloud coverage. On the one hand, as the temporal window increases, the temporal resolution and accuracy of the result will decrease (blur). For example, it has been demonstrated that the accuracies of the 2-, 4-, 6-, and 8-day combinations are 89.5%, 89.0%, 88.2%, and 87.8%, respectively (Gao et al., 2010b). On the other hand, the remaining cloud cover will decrease with the increase of the temporal window span. Hence, it is a trade-off between the remaining cloud and the blurring of the snow information (Hüsler et al., 2014). The longer temporal window also corresponds to a larger area of snow cover. Thus, a balance between temporal resolution and SCA should be considered. When cloud covers the pixel for more than the temporal window, MDC does not work. In this situation, the remaining cloudy pixels should be processed by other methods.

### 3.2.3 Season filter (SFil)

Among the temporal methods, SFil (Gafurov and Bárdossy, 2009;Gafurov et al., 2013;Lindsay et al., 2015) uses the longest time-series information to reclassify the cloudy pixels. This method needs two thresholds in one unbroken hydrological year: complete snowmelt day and snow accumulation start day. As shown in Figure 6, the complete snowmelt day is the day when the pixel is no longer covered by snow, and the snow accumulation start day is the start day when the snow accumulates. The hydrological year is divided into a land season and a snow season by SFil (Da Ronco and De Michele, 2014a). For example, the cloudy pixels before the snow accumulation start day and after the complete snowmelt day are in the land season, and they will be reclassified as land.

   SFil can remove all the cloud cover; however, it does not take the phenomenon of more than one snow cycle occurring in one hydrological year into consideration. Thus, three thresholds are introduced in each hydrological year (Paudel and Andersen, 2011):  the maximum snow extent day, the minimum snow extent day, and snow accumulation start day. In this way, the improved SFil effectively increases the accuracy. Similarly, Lindsay et al. (2015) defined two snow seasons: the full snow season (FSS) and the continuous snow season (CSS). The FSS is the period between the first day and the last day of snow cover, and the CSS has snow cover for at least 14 days, and with intervening snow-free periods of no more than two days. Since the short-term snowfall or snowmelt is not considered by SFil, it is often applied after other spatial or temporal methods. On the whole, SFil obtains a slightly lower accuracy than other temporal filter methods.



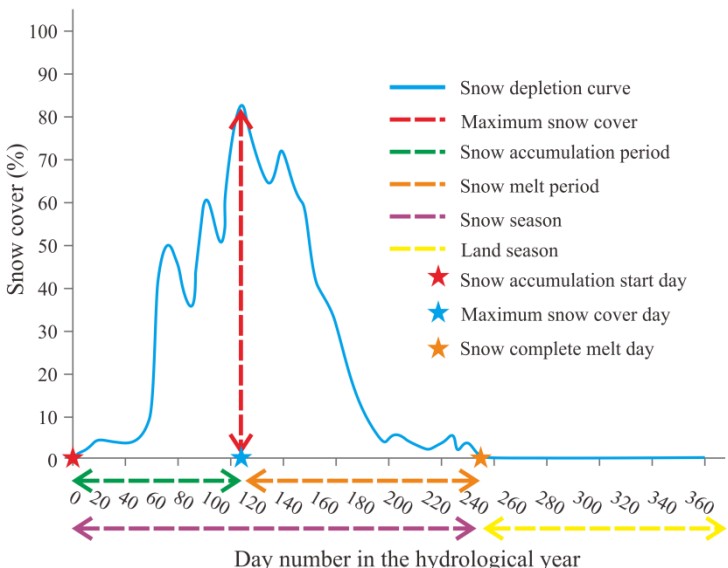

**Figure 6. Snow cover depletion curve and the threshold days.**

### 3.2.4 Temporal interpolation using a mathematical function

As stated previously, ATD, MDC, and SFil are applied to the binary MODIS product, and very little attention has been paid to the FSC product. As an example, Tang et al. (2013) filled the cloud-contaminated pixels by cubic spline interpolation with the cloud-free observation pixels, and the mean absolute error was less than 0.1. This method is based on the rapidly changing and daily shifting features of cloud. In order to distinguish from the temporal interpolation of SCA products, the temporal interpolation for FSC products is called temporal interpolation using a mathematical function, which involves

interpolating the cloudy information of the FSC product along the temporal dimension of the same pixel (Tang et al., 2013;Tang et al., 2017;Xu et al., 2017).

In summary, the temporal methods make use of the cloud instability and the snow correlation in time. They can effectively reduce the cloud cover, partly or completely, and the accuracy is high. ATD and MDC (with not long enough time series) cannot reduce the cloud cover completely and may neglect short snowfall events. SFil can remove the cloud cover

completely. However, when the cloud cover exists persistently in a region, the accuracy of the temporal methods will decrease. For the FSC products, temporal interpolation using a mathematical function is effective (Tang et al., 2017).

### 4. Spatio-temporal methods

Although the spatial methods and temporal methods remove the cloud with a high accuracy, the majority of them cannot reduce the cloud completely. In order to minimize the extent of the cloud cover, many scholars have come up with spatio-





temporal methods to use the complementary advantages of the temporal methods and spatial methods. This type of methods relies on the correlations of snow cover in space and time with two basic forms. One is to utilize the spatial and temporal information step by step, usually as a multi-step combination (MSC) method (Parajka and Blöschl, 2008;Da Ronco and De Michele, 2014b;Gurung et al., 2011;Zhou et al., 2013;Şorman et al., 2019). The other is to utilize the spatial and temporal

information simultaneously (Li et al., 2017;Xia et al., 2012;Huang et al., 2018;Poggio and Gimona, 2015), which we call one-step utilization (OSU).

## 4.1 Multi-step combination (MSC)

As shown in Figure 7, MSC combines the spatial methods and temporal methods alternately (Dariane et al., 2017). Certainly, this is a simple and basic way to exploit the snow cover information in space and time. Although the spatial methods or

temporal methods are not able to get rid of cloud absolutely, their successive combination can make a difference. By removing cloud progressively in each step, an accumulated cloud-free result can be obtained by MSC. In other words, attempts at further reducing the residual cloud in the result of the previous step are made in the next step. Among the combined multiple steps, the TAC method is the first step in most cases. SF, SNOWL, temporal filters, and so on, are then applied in a variety of ways.

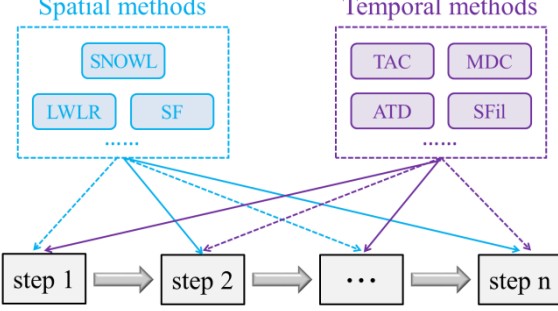

**Figure 7. Multi-step combination.**

For example, a three-step method of TAC, SF, and temporal filter in turn was proposed (Parajka and Blöschl, 2008);

Gafurov and Bárdossy (2009) proposed six steps of TAC, MDC, SNOWL, SF with direct side-border neighboring pixels, SF with eight neighboring pixels, and SFil; Paudel and Andersen (2011) proposed a five-step approach of TAC, ATD, SF with four neighboring pixels, SNOWL, and SFil. López-Burgos et al. (2013) proposed a four-step combination of TAC, temporal filter, SF, and LWLR; and Da Ronco and De Michele (2014a) proposed the five-step method of TAC, conservative temporal filter, SNOWL, six-day backward temporal filter, and SFil.

Whatever the combination of spatial methods and temporal methods, MSC independently utilizes snow correlations in space and time, and the result from previous step directly influences the next step. The combination order of the multiple steps is determined by their respective characteristics. For example, SNOWL requires the CF to be less than 30%, which can





be satisfied after the use of the TAC method and other temporal methods. As a result, SNOWL often follows close behind these methods. No matter what the combination, MSC needs to take a feasible strategy based on the topographic features, temporal variation, and spatial heterogeneity of the snow cover. At the same time, there exists a trade-off between the accuracy and CF for the various steps. These simple MSC methods have been proved to be effective and efficient in cloud

reduction, and agree well with station observations (Paudel and Andersen, 2011). However, this independent and successive utilization cannot take full consideration of the spatio-temporal information.

## 4.2 One-step utilization (OSU)

In contrast, OSU utilizes the spatio-temporal information simultaneously, rather than step by step. As we know, to date, most attention has been paid to the MSC methods. It is only in the last few years that efforts have been made with OSU methods.

The OSU methods are introduced in the following.

Xia et al. (2012) first introduced variational interpolation (Shen and Zhang, 2009) from the image processing field to construct a three-dimensional implicit function with five consecutive daily images (space-time manifold), which has an advantage over representing the complicated surfaces in high-dimensional spaces. The shape of the snow boundary can be easily obtained by the implicit function. It is a good case of utilizing the snow cover evolution with continuity in space and

time. The results indicate that variational interpolation can maintain a close accuracy to the original product. However, its computational efficiency needs to be improved.

The cloud filling (CFil) method (Poggio and Gimona, 2015) is a hybrid of the generalized additive model (GAM) and the geostatistical space-time model. The multi-dimensional spatio-temporal GAM models the binary variables, and geostatistical kriging accounts for the spatial details. The space-time correlations of snow cover are well exploited by the CFil method.

Even for a high fraction of cloud cover, the CFil method still achieves a satisfactory accuracy and is suitable for the seasonal variation of snow cover. However, the requirement for a lot of ancillary data (e.g., land surface temperature (LST), land cover, and soil pattern data) limits its application, to some degree.

The adaptive spatio-temporal weighted method (ASTWM) (Li et al., 2017) estimates the cloudy pixel according to the probability of snow cover, which is the adaptively weighted combination of the snow probabilities in space and time.

Experiments demonstrated that ASTWM not only removes the cloud completely, but also achieves a high OA of above 93% under different CFs. However, ASTWM resorts to the optimal weight of spatial and temporal probability with a high cost, and it may be able to reclassify snow cover as not snow cover under darker conditions (Krajčí et al., 2014).

Combining the spatio-temporal-spectral information and environmental relation, Huang et al. (2018) proposed a hidden Markov random field (HMRF) based spatio-temporal model. The information of the cubic spatio-temporal neighborhood is

effectively utilized by the HMRF-based method. The snow mapping accuracy of HMRF-based method can achieve 88% and decrease the cloud cover to 1%, which improves the overall snow mapping accuracy and reduces the omission error of the original product. For the challenging task of snow cover mapping in the transition periods of snow and in forest areas, it also obtains promising results.



Recently, more and more spatio-temporal methods have been proposed to alleviate cloud cover. In fact, MSC has a numerical advantage over OSU. MSC usually makes an assurance of removing all the cloud by successive multiple steps. In contrast, OSU achieves this goal by one step. OSU jointly utilizes the spatio-temporal information from the snow coverage, which contributes to more promising cloud removal results.

**5. Multi-source fusion methods**

The above-mentioned three types of methods mainly use the spatial and temporal information of the snow cover from the same optical remote sensing sensor. In contrast, multi-source fusion methods (Romanov et al., 2000;Gao et al., 2010a;Yu et al., 2012;Gafurov et al., 2015;Wang et al., 2015;Dong and Menzel, 2016) utilize the complementary information among different sources (Shen et al., 2016), such as optical observations, microwave observations, station observations, and so on,

as shown in Figure 8.

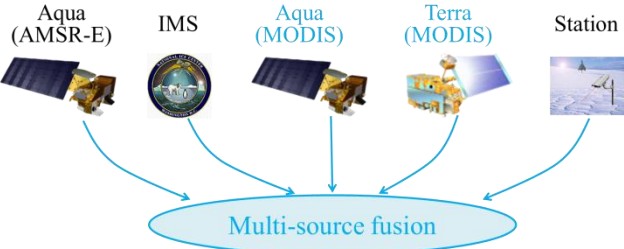

**Figure 8. Multi-source fusion.**

**5.1 Optical and microwave observations**

In general, optical-based products have a high spatial resolution and are influenced by cloud cover, whereas microwave-based products have a low resolution and a good cloud-penetrating capacity. Therefore, the fusion of optical and microwave data has been the most representative multi-source fusion method of cloud removal, e.g., MODIS and AMSR-E (Liang et al., 2008a;Gao et al., 2011b;Akyurek et al., 2010;Huang et al., 2014;Deng et al., 2015;Li et al., 2014a;Bergeron et al.,

2014;Huang et al., 2016), GOES and SSM/I (Romanov et al., 2000), and the visible/infrared spin-scan radiometer (VISSR) and microwave radiation imager (MWRI) (Yang et al., 2014).

The fusion of MODIS and AMSR-E is the most frequently used method. For the AMSR-E SWE product, a value of zero represents a land surface, and values of 1–240 represent a snow-covered surface. The cloudy pixels of the MODIS product are reclassified as land if SWE=0, or as snow if SWE ranges from 1 to 240 (Wang et al., 2015). For more fusion rules, please

refer to (Liang et al., 2008a). After recoding the MODIS and AMSR-E products, some scholars have adopted a rule of maximum value composite (MAC) (Yu et al., 2012;Wang et al., 2018). It is noted that the spatial resolution (25km) of AMSR-E SWE should be resampled to the 500m of the MODIS snow cover product before fusion. In addition, for AMSR-E



SWE, the gaps with daily changing locations are filled by the ATD method (Wang et al., 2015). The fusion accuracy is much higher than the MODIS daily and eight-day products (Liang et al., 2008a). However, the fusion decreases the spatial resolution and results in uncertainties, because of the low spatial resolution of AMSR-E (Gao et al., 2010b;Gao et al., 2010a).

In addition, to remove the cloud cover in the MODIS product, Yu et al. (2016) developed the method of fusing MODIS and
the combined IMS product, where the cloudy pixels of MODIS are replaced by the values of the IMS pixels. As stated previously, the IMS product itself is a combination of optical and microwave data. Compared with the AMSR-E spatial resolution of 25km, the cloud-free IMS product has a much higher spatial resolution of ~1km and ~4km, which is also resampled to the same resolution as MODIS. Therefore, the fusion of MODIS and the IMS product has an advantage over the fusion of MODIS and AMSR-E. Moreover, Wang et al. (2018) proposed to fuse MODIS and the AMSR-E daily SWE
product or IMS product. They pointed out that the accuracy of the cloudy pixel reclassification is dependent on AMSR-E and IMS.

## 5.2 Optical and station observations

The observations of the existing meteorological stations are long-term, high-precision, and point-based. In contrast, remote sensing observations are spatially continuous. Based on the correlations between station observations and spatial snow cover
patterns, the snow cover can be reconstructed by fusing station data and high spatial resolution remote sensing data. For example, based on station observations and optical observation data, the conditional probability (CP) of each cloudy pixel being snow can be calculated to reclassify the residual cloudy pixels (Dong and Menzel, 2016;Gafurov et al., 2015;Gafurov et al., 2016). The CP represents the probability of a pixel being snow cover, on the condition of the SD being higher than zero at the station. The CP also implies the similarity between different meteorological stations. In other words, the snow
presence in one station is predicted by the information about the presence of snow at another station. This method has been confirmed to be effective, especially during the snow season. In the Zerafshan River basin, the accuracy is only slightly lower than the original MODIS product (Gafurov et al., 2015), according to Landsat-derived snow cover (Gafurov et al., 2013). However, to some degree, the distribution and number of meteorological stations limits the predictive ability of snow cover reconstruction.

Gafurov et al. (2016) developed an all-in-one software package called MODSNOW-Tool, with advanced cloud removal algorithms for MODIS snow cover products. The integrated algorithms include the six-step method in (Gafurov and Bárdossy, 2009) and the CP method in (Gafurov et al., 2015). MODSNOW-Tool is equipped with operational and non-operational modes, which consist of seven processing modules. The operational mode generates a daily cloudless snow cover map without user interaction, and the non-operational mode generates a historical snow cover map. This tool can remove the
complete cloud cover, which is a major breakthrough.



### 5.3 Optical, microwave, and station observations

In terms of multi-source fusion methods, the fusion of optical and microwave data is the most common approach, and the fusion of optical and station observations has attracted relatively little attention. The moderate attention has been paid to the fusion of optical, microwave, and station observations. Brown et al. (2010) utilized 10 data sources, including optical,

microwave, and station observations, to estimate the cloud-free snow cover. Before the fusion of the 10 data sources, the consistency of each dataset was assessed by their correlations. The fusion result was then obtained by converting the average anomaly series to the first differences then joint the difference series, in which the average anomaly series was computed from each reference period (Brown et al., 2010). This method has been well validated for monitoring the SCE variation in the Arctic region (Brown et al., 2010).

### 6. Validation and evaluation

### 6.1 In-situ and remote sensing data based evaluation

The evaluation of the accuracy and effectiveness of cloud removal is also very important to the spatio-temporally continuous MODIS snow cover products. When the SD data of climate stations are available, a time series of in-situ observations can be used to validate the temporal effectiveness of the cloud-removed products in real-data

experiments. The SD data of the in-situ observations are usually set as the standard data. The nearest pixel to the site is classified as snow when the SD exceeds a threshold value; otherwise, it is classified as no snow (Parajka and Blöschl, 2008). Through the comparison of the in-situ SD and the reclassified snow cover product, the effect can be evaluated.

In-situ observation based evaluation is a direct and valid method. However, in most cases, because of data

privacy policies or the absence of climate stations, researchers cannot acquire in-situ observations of snow cover. To conduct the validation, the majority of researchers resort to remote sensing data based evaluation via simulated experiments, following the work of Gafurov and Bárdossy (2009). Firstly, the Aqua and Terra snow cover products were combined by the TAC method. A number of images with CF of less than 10% were selected as "truth" products, and the cloud masks of the other images with a larger CF were applied to cover the "truth"

products and get the "observation" products. Next, the "observation" products were reclassified. Finally, the results were compared with the "truth" products. In addition, some other SCA products with a higher resolution can also be considered as the true ground data. For example, the Landsat TM/ETM+/OLI FSC products have been used to validate the MODIS FSC products (Liang et al., 2017;Crawford, 2015;Kuter et al., 2018). However, when the MODIS snow products are covered by cloud, the optical-based products with a higher spatial resolution

will also be covered by cloud. This method is thus only effective for the validation of clear-sky accuracy.





## 6.2 Quantitative evaluation indicators

There have been many kinds of indicators used to evaluate the result of generated spatio-temporally continuous snow products. Usually, the effectiveness is described by CF, while the accuracy is evaluated by OA, overall clear-sky accuracy (OC), overestimation error (OE) and underestimation error (UE) (Gafurov and Bárdossy, 2009), which are calculated by the following expressions:

$$OA = \frac{N_s^s + N_{ns}^{ns}}{N_a} \tag{1}$$

$$OC = \frac{N_s^s + N_{ns}^{ns}}{N_c} \tag{2}$$

$$OE = \frac{N_s^{ns}}{N_c} \tag{3}$$

$$UE = \frac{N_{ns}^s}{N_c} \tag{4}$$

where $N_a$ denotes the amount of cloudy pixels from the observation product; $N_c$ denotes the amount of cloudy pixels in the observation product; $N_s^s$ denotes the amount of cloudy pixels which are reclassified as snow pixels in the observation product and are snow pixels in the true product; $N_{ns}^{ns}$ denotes the amount of cloudy pixels which are reclassified as not snow pixels in the observation product and are not snow pixels in the true product; $N_s^{ns}$ denotes the amount of cloudy pixels which are reclassified as snow pixels in the observation product and are not snow pixels in the true product; and $N_{ns}^s$ denotes the amount of cloudy pixels which are reclassified as not snow pixels in the observation product and are snow pixels in the true product.

## 7. Future directions

The most common algorithms of cloud removal for MODIS snow products have been aimed at the binary product (V005). In the spring of 2016, MODIS Collection 6 (C6) products were published (Malmros et al., 2018). In the MODIS C6 products, the binary SCA products have been substituted by the NDSI, and the FSC product is not supplied at all (Hall and Riggs, 2016a, b;Riggs et al., 2017). Research work has demonstrated that the Terra MODIS C6 FSC has a strong spatial and temporal agreement with Landsat TM/ETM+ (Kuter et al., 2018). More and more researchers are thus moving to MODIS C6 data (Malmros et al., 2018;Huang et al., 2018).

On the one hand, the aforementioned spatial methods, temporal methods, spatio-temporal methods, and multi-source fusion methods will still work well, on the condition that the newly released MODIS C6 data are converted into a binary





SCA product by the MODIS snow mapping algorithm SNOMAP. On the other hand, the cloud removal algorithms for the NDSI products should also be paid attention to. For the snow product transformation from binary product to NDSI, the cloud removal methods should be changed accordingly. The NDSI has the property of seasonal periodicity, and is similar to a vegetation index (VI), to some degree, e.g., the normalized difference vegetation index (NDVI) or enhanced VI. In this sense,

the methods of reconstructing a gap-free VI (Yang et al., 2015;Chen et al., 2004;Lovell and Graetz, 2001;Roerink et al., 2000;Poggio et al., 2012) will provide a reference for the cloud-free NDSI. Without doubt, the particular features of the NDSI should be considered.

In terms of the cloud removal algorithms for the NDSI, machine learning (Tahsin et al., 2017), with its advantages in processing high-dimensional data, will also be a promising direction. Owing to the long time series of observed snow cover,

the variation rule of snow cover could be discovered more easily by machine learning. Moreover, with the rapid technical development of unmanned aerial vehicles (UAVs), UAVs will act as an effective supplement for snow mapping. For example, Liang et al. (2017) successfully applied UAV for snow coverage mapping of Tibetan Plateau. We therefore predict that UAVs will play a significant part in the cloud removal of MODIS snow cover products in the near future.

## 8. Conclusions

In this paper, the existing methods of generating spatio-temporally continuous snow cover products have been summarized from the four aspects of spatial algorithms, temporal algorithms, spatio-temporal algorithms, and multi-source fusion algorithms. These methods utilize the spatial and temporal variation characteristics and the complementary properties of different observation approaches. Thanks to the spatially correlated relations of snow cover, the spatial methods are relatively effective in the removal of neighboring cloudy pixels, but are usually powerless for large-area cloud cover. The

temporal methods remove the cloud cover of the products using the temporal variation rule of snow cover. In addition to a high accuracy of cloud removal, the temporal methods have the ability to remove all the cloud, on the condition that the time series is long enough. As their name implies, the spatio-temporal methods take advantage of the spatial methods and temporal methods by successive or one-step utilization of them. The multi-source fusion methods are based on the complementary observations of different types. The fusion of optical, microwave, and station observations contributes to a

promising cloud removal result.

## Acknowledgments

This work was supported by the National Natural Science Foundation of China (NSFC) under grant no. 41701394; the Hubei Natural Science Foundation under grant no. 2017CFB189; the Open Research Fund of the Key Laboratory of Spatial Data Mining & Information Sharing of Ministry of Education, Fuzhou University, under grant no. 2018LSDMIS02; the Key

Laboratory of Satellite Mapping Technology and Application, the National Administration of Surveying, Mapping and Geoinformation, under grant no. KLSMTA-201703; the Key Laboratory of Digital Earth Science, the Institute of Remote Sensing and Digital Earth, the Chinese Academy of Sciences, under grant no. 2016LDE004; and the Fundamental Research



Funds for the Central Universities under grant no. 2042017kf0034. The authors would also like to thank the anonymous reviewers.

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

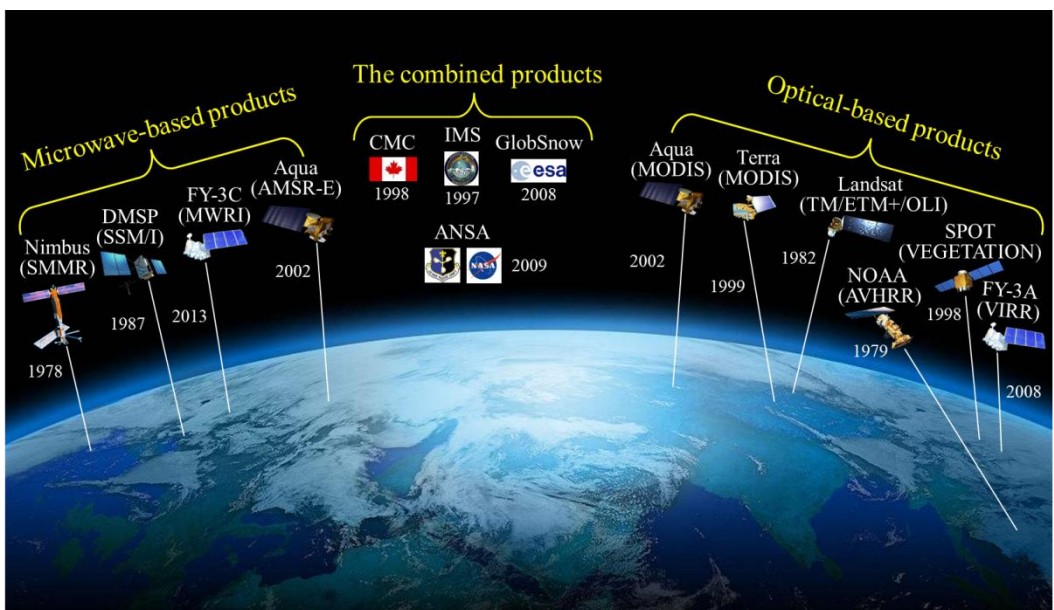

**Figure 1**. **Remote sensing based snow cover products.**

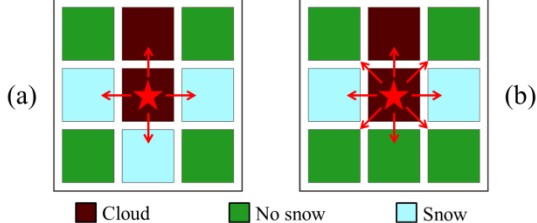

**Figure 9. Spatial filter. (a) Four-pixel neighborhood. (b) Eight-pixel neighborhood.**



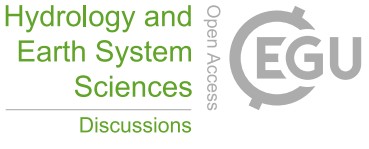

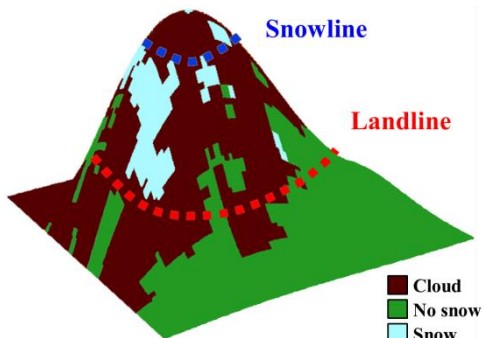

Figure 3. Snowline and landline.

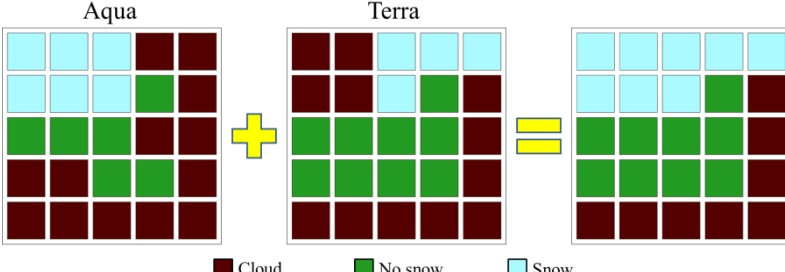

5    Figure 4. Terra and Aqua combined.

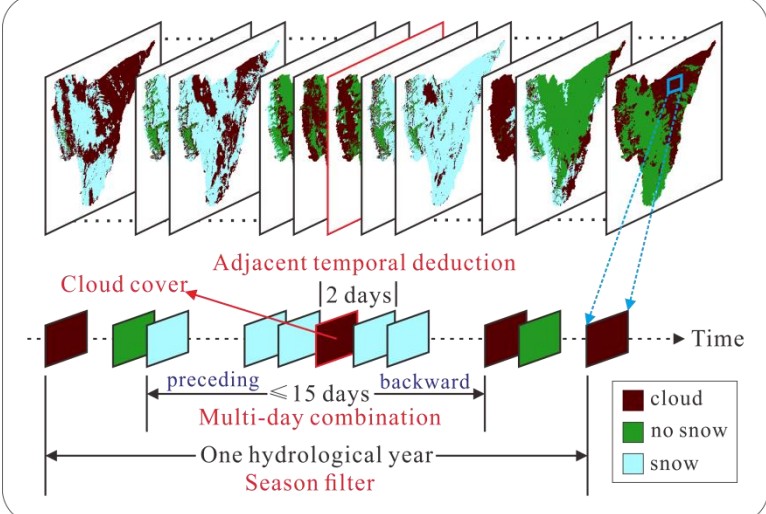

Figure 5. Temporal filters for the MODIS SCA products.



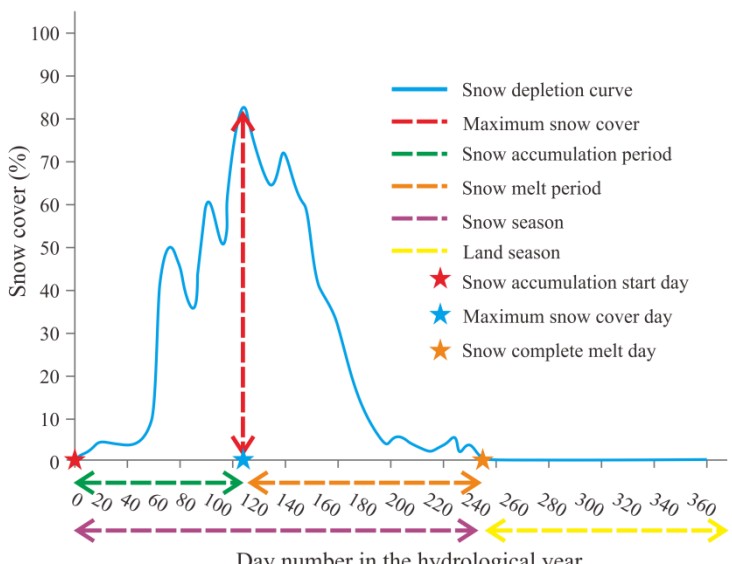

**Figure 6. Snow cover depletion curve and the threshold days.**

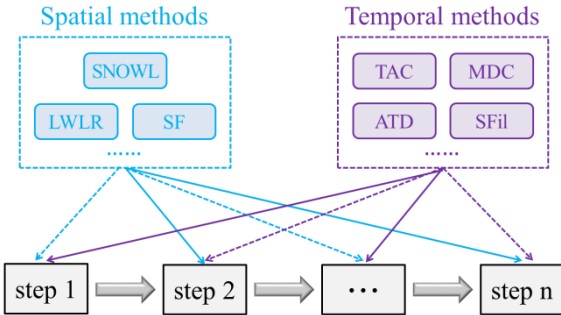

5    **Figure 7. Multi-step combination.**

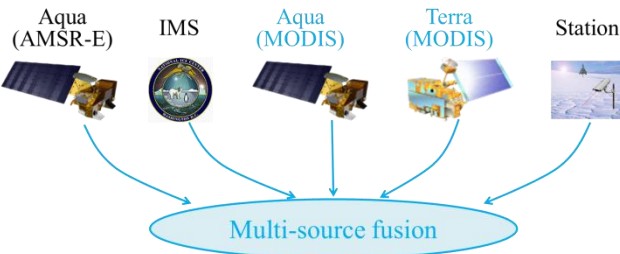

**Figure 8. Multi-source fusion.**

