# Peer review of "The recent developments in cloud removal approaches of MODIS snow cover product"

_Hydrology and Earth System Sciences, 2018_

## Referee Comment (RC1) · Anonymous Referee #1 · 3 Apr 2019

Li et al review the state of the art of cloud-removal procedures for satellite optical snow products. They focus on MODIS products as these are among the most popular datasets used in the community. They classify existing approaches as spatial, temporal, spatio-temporal, and multi-source methods (that is, methods relying on more than one sensor/platform). They include a brief future-direction section that is mostly geared toward MODIS version 6 and the potential role of machine learning and UAVs for cloud removal.

Several cloud-removal procedures have been recently proposed, which significantly enhanced the applicability of satellite optical products for snow science. From this standpoint, I believe that a review of existing methods is timely and could be an important contribution for the readers of HESS. I also appreciated the effort of authors

to summarize many diverse algorithms in a relatively brief and well-structured paper. That said, there are several points that the authors may want to address. Some of them regard the overall content of the manuscript, while some others regard specific passages. I summarize both below.

The most pressing remark (to me) concerns the future-direction section. According to HESS guidelines, "review articles summarize the status of knowledge and outline future directions of research within the journal scope". I think that the manuscript is extensive in terms of synthesis of the state of the art, and some details may be even summarized. On the other hand, the manuscript is very brief in terms of future directions and in general in terms of research needs and knowledge gaps, which should to me be as central for a review as the state of the art. In other words, what are the main scientific knowledge gaps that authors see in this field and that should be addressed in the future? Is there any unexplored hypothesis related to cloud-removal procedures that authors would like to point out for future research? For example, how could new satellites with a higher resolution than MODIS change this field? Also, what is the specific role that authors envision for UAVs, as these sensors have been generally applied to small patches and are not (to my knowledge) deployed operationally? The latter would guarantee the short revisiting time needed by a cloud-removal procedure. I believe that the manuscript may benefit from more details about these points (some details are already scattered throughout the text) and extensive summarization of the technical details of each cloud-removal procedure.

Some wording choices may also be reconsidered. For example, the manuscript uses a quite extensive number of acronyms and abbreviations that made my reading of specific passages quite difficult. I suggest authors limit acronyms to those that are well known in the community and avoid acronyms that are used only a few times in the text. I point to other examples of wording choices below.

I finally found figures to play a quite marginal role in the manuscript as it is now. For example, I think that Figures 2, 4, 7, and 8 do not add much information to what is

currently written in the corresponding Sections (especially the example in Figure 4 is easy to understand even without the figure: the so-called TAC method simply blends maps from Aqua and Terra satellites). Figure 3 is also quite confusing to me. Maybe more examples of real-world results from previous papers may make a more effective point and could also serve as a basis for commenting limitations and future directions of research?

SPECIFIC – MINOR COMMENTS

- Title: maybe mention MODIS?

- Abstract: please avoid "and so on" here and throughout the text: it would be more informative for readers to include all items in a list that are deemed essential to understand the concept.

- Line 11 page 1: please be more specific on what is the temporal scale of "discontinuity for long-term monitoring".

- Line 15 -16 page 1: please revise the sentence starting with "Experiments were conducted..." as it sounded to me as if these experiments were conducted within this paper. Also, consider including some takeaways about the most important knowledge gaps that authors see in the field.

- Line 21 page 1: more and more -> increasingly (here and throughout the text).

- Line 24 page 1: the short revisit period is relative: for example, for flood control or other emergency situations one would ideally need sub-daily, or even sub-hourly snapshots of snow distribution.

- Line 27 page 1: it is a bit confusing that you first discuss combined products and then optical products (here and in the following paragraphs) as the latter are one of the data sources for the first.

- Line 12ff page 2: please revise qualitative terms like "quickly cover", "high temporal

resolution", "so low", "high spatial resolution" with more quantitative terms.

- Line 19 page 2: please include some more details about the "limitations of the imaging orbit" for readers that are less familiar with this field.

- Line 19 page 3: please include some examples of these problems here.

- Line 21 page 3: again, please replace "poor" with more quantitative terms.

- Line 25 page 3: please replace "higher spatial resolution" with more quantitative terms.

- Line 32 page 3: please replace "long observation time series" and "high spatial resolution" with more quantitative terms.

- Line 8ff page 4: please include some examples of other products (and use quantitative terms) to make these advantages clearer. Also, consider introducing in this paragraph v6 as well.

- Line 5 page 5: maybe change "spatial distribution property" with "spatial patterns"?

- Line 3 page 6: is there any way to quantify "not very sensitive" here?

- Line 20 page 6: carry out -> implement.

- Line 24 page 6: what do you mean with "In a few special cases, the snowline for the whole area is not met"?

- Line 26 page 6: maybe define "commission error" for readers that are less familiar with this field.

- Line 3 page 7: what do you mean with "LWLR enforces on two explanatory variables"?

- Line 7 page 7: please quantify the "high cost".

- Line 21 page 7: changeful -> very variable.

- Line 23 page 7: what do you mean with "obvious accuracy"?
- Line 4 page 8: maybe complete snowmelt? Partial snowmelt would not, in my opinion, be an issue here.

- Line 7 page 8: please replace " in the early days" with a more specific time period.

- Line 9 page 8: what are these slight differences and under which specific circumstances do they emerge?

- Line 19 page 8: please cite some of this literature here.

- Line 16 page 9: it is not clear to me why the fact that accuracy will be lower during transitional periods make this method unsuitable for operations.

- Line 20 page 9: please be more specific with regard to the "constant or flexible way", for example by mentioning some examples.

- Line 9 page 12: what do you mean with "a simple and basic way to exploit the snow cover information"? also, please consider revising wording like "get rid of" or "and so on".

- Line 1 page 13: "As a result …... Methods" -> "As a result, SNOWL is often applied immediately after these methods".

- Line 16 page 13: please provide some more information in support of this final sentence.

- Line 23 page 14: please include units (mm?) for SWE

- Line 19 page 18: powerless -> less effective (or simply ineffective)

---

## Referee Comment (RC2) · Anonymous Referee #2 · 16 Apr 2019

The recent developments in spatio-temporally continuous snow cover product generation.

By: Li et al.

This study reviews the existing methodologies of cloud removal from optical remote sensing based snow cover data. The focus is given to MODIS snow cover data, which is becoming valuable due to prolonged time series. There has been published several methods focusing on cloud elimination from MODIS data and this manuscript summarizes those existing approaches very well. Thus, it can be of interest to a wider audience interested in snow research.

The study was initially submitted to WRR, where I was assigned as a reviewer as well.

The comments given then are well considered in this version of the manuscript. The review of existing methods on cloud removal is well structured and easy to read. Only, the connection to UAV in the chapter "future directions" may be irrelevant as they do not provide continuous observation of snow cover that can be used for cloud removal. Moreover, UAVs and satellites observe at different scales, which can be hardly combined. Rather, I suggest to include some text about the potential of Sentinel product in providing snow cover data with higher spatial resolution in the future.

Considering my previous comments and significant improvement of the manuscript, I suggest it for publication in HESS after a minor revision, considering comments below.

General comments:

1. The title of the manuscript does not really reflect the content. The manuscript is about cloud removal approaches and not about the generation of continuous snow cover product. Also, authors can specifically mention the word MODIS in the title as most (if not all) methods use MODIS data for cloud removal.

2. The authors are encouraged to address some words about the potential application of Sentinel product in generating snow cover maps and its advantages and drawbacks with regard to MODIS. This can be discussed in the "future directions" and "conclusions and discussion" chapter.

Technical comments:

1. Insert a tab separation while listing references in brackets in the text.
* * *

---

## Author Comment (AC1) · 18 Apr 2019

**Response to comments by Reviewer #2:**

We would like to take this opportunity to gratefully thank the reviewer for his/her constructive comments and recommendations. An item-by-item, point-by-point response to the interesting comments raised by the reviewer follows.

1. **This study reviews the existing methodologies of cloud removal from optical remote sensing based snow cover data. The focus is given to MODIS snow cover data, which is becoming valuable due to prolonged time series. There has been published several methods focusing on cloud elimination from MODIS data and this manuscript summarizes those existing approaches very well. Thus, it can be of interest to a wider audience interested in snow research. The study was initially submitted to WRR, where I was assigned as a reviewer as well. The comments given then are well considered in this version of the manuscript. The review of existing methods on cloud removal is well structured and easy to read. Only, the connection to UAV in the chapter "future directions" may be irrelevant as they do not provide continuous observation of snow cover that can be used for cloud removal. Moreover, UAVs and satellites observe at different scales, which can be hardly combined. Rather, I suggest to include some text about the potential of Sentinel product in providing snow cover data with higher spatial resolution in the future. Considering my previous comments and significant improvement of the manuscript, I suggest it for publication in HESS after a minor revision, considering comments below.**

*Response:* We are very grateful to the Reviewer for taking your valuable time to read this manuscript. We truly appreciate this chance to gain your insight and views on these issues. Thank you very much for your comments. According to your suggestions, we've amended the relevant parts in the revised manuscript. The introduction of Sentinel has been added to our revised version.

2. **The title of the manuscript does not really reflect the content. The manuscript is about cloud removal approaches and not about the generation of continuous snow cover product. Also, authors can specifically mention the word MODIS in the title as most (if not all) methods use MODIS data for cloud removal.**

*Response:* Thanks for your suggestion. We have changed the title into "The recent developments in cloud removal approaches of MODIS snow cover product".

**3. The authors are encouraged to address some words about the potential application of Sentinel product in generating snow cover maps and its advantages and drawbacks with regard to MODIS. This can be discussed in the "future directions" and "conclusions and discussion" chapter.**

*Response:* Thanks for your constructive comments. We have added the description on Sentinel product to the future direction. In the framework of the multi-source fusion, the microwave-based observation with a higher spatial resolution than AMSR-E should make a difference, especially the Sentinel series. For example, Sentinel-1 SAR is with the spatial resolution of 20m, which will significantly improve the fusion accuracy of MODIS snow cover product. Additionally, the optical observations of Sentinel series, e.g., Sertinel-2 Multispectral Instrument (MSI) and Sentinel-3 Sea Land Surface Temperature Radiometer (SLSTR), also have the potential in providing snow cover product with higher spatial resolution in the future.

**4. Insert a tab separation while listing references in brackets in the text.**

*Response:* We have inserted a blank space into the reference lists in the brackets.

Last but not least, we gratefully thank the reviewer again for his/her very interesting comments and suggestions, which greatly helped us to improve the technical quality and presentation of our manuscript.

---

## Author Comment (AC2) · 18 Apr 2019

**Response to comments by Reviewer #1:**

We would like to take this opportunity to gratefully thank the reviewer for his/her constructive comments and recommendations. An item-by-item, point-by-point response to the interesting comments raised by the reviewer follows.

1. **Li et al review the state of the art of cloud-removal procedures for satellite optical snow products. They focus on MODIS products as these are among the most popular datasets used in the community. They classify existing approaches as spatial, temporal, spatio-temporal, and multi-source methods (that is, methods relying on more than one sensor/platform). They include a brief future-direction section that is mostly geared toward MODIS version 6 and the potential role of machine learning and UAVs for cloud removal. Several cloud-removal procedures have been recently proposed, which significantly enhanced the applicability of satellite optical products for snow science. From this standpoint, I believe that a review of existing methods is timely and could be an important contribution for the readers of HESS. I also appreciated the effort of authors to summarize many diverse algorithms in a relatively brief and well-structured paper. That said, there are several points that the authors may want to address. Some of them regard the overall content of the manuscript, while some others regard specific passages. I summarize both below.**

*Response:* We are very grateful to the Reviewer for taking your valuable time to read this manuscript. We truly appreciate this chance to gain your insight and views on these issues. Thank you very much for your comments. According to your suggestions, we've amended the relevant parts in the revised manuscript.

2. **The most pressing remark (to me) concerns the future-direction section. According to HESS guidelines, "review articles summarize the status of knowledge and outline future directions of research within the journal scope". I think that the manuscript is extensive in terms of synthesis of the state of the art, and some details may be even summarized. On the other hand, the manuscript is very brief in terms of future directions and in general in terms of research needs and knowledge gaps, which should to me be as central for a review as the state of the art. In other words, what are the main scientific knowledge gaps that authors see in this field and that should be addressed in the future? Is there any unexplored hypothesis related to cloud-removal procedures that authors would like to point out for future research?**

**For example, how could new satellites with a higher resolution than MODIS change this field? Also, what is the specific role that authors envision for UAVs, as these sensors have been generally applied to small patches and are not (to my knowledge) deployed operationally? The latter would guarantee the short revisiting time needed by a cloud-removal procedure. I believe that the manuscript may benefit from more details about these points (some details are already scattered throughout the text) and extensive summarization of the technical details of each cloud-removal procedure.**

*Response:* Thanks for your constructive suggestion. The future directions are reordered in the revised version. On the whole, multi-source fusion is still the promising direction of the cloud removal algorithm of MODIS snow cover product. The higher spatial resolution snow cover product from Sentinel will play a more important role in the cloud removal mission. UAVs will work on the accuracy validation of the cloud-removed MODIS snow cover products in the near future. Additionally, the new algorithms for MODIS Collection 6 (C6) products should be developed correspondingly.

3. **Some wording choices may also be reconsidered. For example, the manuscript uses a quite extensive number of acronyms and abbreviations that made my reading of specific passages quite difficult. I suggest authors limit acronyms to those that are well known in the community and avoid acronyms that are used only a few times in the text. I point to other examples of wording choices below.**

*Response:* We have deleted some abbreviations as the reviewer suggested.

4. **I finally found figures to play a quite marginal role in the manuscript as it is now. For example, I think that Figures 2, 4, 7, and 8 do not add much information to what is currently written in the corresponding Sections (especially the example in Figure 4 is easy to understand even without the figure: the so-called TAC method simply blends maps from Aqua and Terra satellites). Figure 3 is also quite confusing to me. Maybe more examples of real-world results from previous papers may make a more effective point and could also serve as a basis for commenting limitations and future directions of research?**

*Response:* Figures 2, 3, 4, 7 and 8 are all deleted in the new version. Thanks very much for your advice.

5. **Title: maybe mention MODIS?**

*Response:* Thanks for your suggestion. We have changed the title into "The recent developments in cloud removal approaches of MODIS snow cover product".

**6.** **Abstract: please avoid "and so on" here and throughout the text: it would be more informative for readers to include all items in a list that are deemed essential to understand the concept.**

*Response:* We have deleted it in the abstract and throughout the manuscript.

**7.** **Line 11 page 1: please be more specific on what is the temporal scale of "discontinuity for long-term monitoring".**

*Response:* We have stated that it is for MODIS, so the temporal scale is daily.

**8.** **Line 15 -16 page 1: please revise the sentence starting with "Experiments were conducted. . ." as it sounded to me as if these experiments were conducted within this paper. Also, consider including some takeaways about the most important knowledge gaps that authors see in the field.**

*Response:* Yes, we have deleted this sentence.

**9.** **Line 21 page 1: more and more -> increasingly (here and throughout the text).**

*Response:* Yes, we have done it.

**10.** **Line 24 page 1: the short revisit period is relative: for example, for flood control or other emergency situations one would ideally need sub-daily, or even sub-hourly snapshots of snow distribution.**

*Response:* As suggested, "relatively" is added to the description.

**11.** **Line 27 page 1: it is a bit confusing that you first discuss combined products and then optical products (here and in the following paragraphs) as the latter are one of the data sources for the first.**

*Response:* We have reordered the introduction of snow cover products with the order of microwave-based products, optical-based products and combined products.

**12.** **Line 12ff page 2: please revise qualitative terms like "quickly cover", "high temporal resolution", "so low", "high spatial resolution" with more quantitative terms.**

*Response:* The sentences are modified according to the suggestions. "quickly cover" and "high temporal resolution" are restricted by 3-5 days. "so low" means the spatial resolution >1km. "high spatial resolution" means the spatial resolution is <1km.

**13. Line 19 page 2: please include some more details about the "limitations of the imaging orbit" for readers that are less familiar with this field.**

*Response:* Thanks very much for your suggestion. Due to the imaging orbit gap, microwave-based products are subject to spatial gaps.

**14. Line 19 page 3: please include some examples of these problems here.**

*Response:* We have altered it. Several problems encountered when the component products are used alone, including cloud cover and low accuracy, have been solved.

**15. Line 21 page 3: again, please replace "poor" with more quantitative terms.**

*Response:* "poor" means the spatial resolution >1km.

**16. Line 25 page 3: please replace "higher spatial resolution" with more quantitative terms.**

*Response:* This sentence was deleted in the revised version.

**17. Line 32 page 3: please replace "long observation time series" and "high spatial resolution" with more quantitative terms.**

*Response:* "long observation time series" is from 1960, and "high spatial resolution" means the spatial resolution ≤1km.

**18. Line 8ff page 4: please include some examples of other products (and use quantitative terms) to make these advantages clearer. Also, consider introducing in this paragraph v6 as well.**

*Response:* AVHRR, VEGETATION and VIRR are included here. Since V6 is introduced in the "Future directions", it was not introduced in this paragraph.

**19. Line 5 page 5: maybe change "spatial distribution property" with "spatial patterns"?**

*Response:* Yes, we have done it.

**20. Line 3 page 6: is there any way to quantify "not very sensitive" here?**

*Response:* Thanks very much for your suggestion. However, it is very hard to quantify the sensitivity.

**21. Line 20 page 6: carry out -> implement.**

*Response:* Yes.

**22. Line 24 page 6: what do you mean with "In a few special cases, the snowline for the whole area is not met"?**

*Response:* We mean the snowline for the whole area is very hard to find. We have modified the sentence.

**23. Line 26 page 6: maybe define "commission error" for readers that are less familiar with this field.**

*Response:* It was altered by misclassification error.

**24. Line 3 page 7: what do you mean with "LWLR enforces on two explanatory variables"?**

*Response:* We mean that LWLR uses two variables, and it is altered.

**25. Line 7 page 7: please quantify the "high cost".**

*Response:* The time cost is larger than 20h in the Salt River basin in Arizona.

**26. Line 21 page 7: changeful -> very variable.**

*Response:* Yes.

**27. Line 23 page 7: what do you mean with "obvious accuracy"?**

*Response:* We mean the accuracy is not significantly reduced.

**28. Line 4 page 8: maybe complete snowmelt? Partial snowmelt would not, in my opinion, be an issue here.**

*Response:* Yes, it's complete snowmelt.

**29. Line 7 page 8: please replace " in the early days" with a more specific time period.**

*Response:* "The early days" means before 2012.

**30. Line 9 page 8: what are these slight differences and under which specific circumstances do they emerge?**

*Response:* Since the quantitative image restoration method cannot reconstruct the Aqua MODIS band 6 with the accuracy of 100%, the slight differences still exist.

**31. Line 19 page 8: please cite some of this literature here.**

*Response:* Yes.

**32. Line 16 page 9: it is not clear to me why the fact that accuracy will be lower during transitional periods make this method unsuitable for operations.**

*Response:* In fact, we want to say that ATD is not suitable for a context with variable snow covers.

**33. Line 20 page 9: please be more specific with regard to the "constant or flexible way", for example by mentioning some examples.**

*Response:* Yes, we have added the examples. Given a temporal window of ten days, the constant way of MDC means the combination is implemented in ten days, and the flexible way represents the combination can be implemented in a varying days (≤10 days).

**34. Line 9 page 12: what do you mean with "a simple and basic way to exploit the snow cover information"? also, please consider revising wording like "get rid of" or "and so on".**

*Response:* This sentence was deleted. "get rid of" and "and so on" are replaced by other words.

**35. Line 1 page 13: "As a result . . .. Methods" -> "As a result, SNOWL is often applied immediately after these methods".**

*Response:* Yes, we have done it.

**36. Line 16 page 13: please provide some more information in support of this final sentence.**

*Response:* We have altered the sentence. Its computational efficiency needs to be improved since it intends to retrieve the space-time surface of snow cover on consecutive days.

**37. Line 23 page 14: please include units (mm?) for SWE**

*Response:* Yes.

**38. Line 19 page 18: powerless -> less effective (or simply ineffective)**

*Response:* Yes.

Last but not least, we gratefully thank the reviewer again for his/her very interesting comments and suggestions, which greatly helped us to improve the technical quality and presentation of our manuscript.

---

## Author Comment (AC4) · 18 Apr 2019

The attached file is the revised manuscript.

Please also note the supplement to this comment:
https://www.hydrol-earth-syst-sci-discuss.net/hess-2018-633/hess-2018-633-AC4-supplement.pdf
* * *